# A Novel Concept of Transperineal Focused Ultrasound Transducer for Prostate Cancer Local Deep Hyperthermia Treatments

**DOI:** 10.3390/cancers15010163

**Published:** 2022-12-27

**Authors:** Pauline Coralie Guillemin, David Sinden, Yacine M’Rad, Michael Schwenke, Jennifer Le Guevelou, Johannes W. E. Uiterwijk, Orane Lorton, Max Scheffler, Pierre-Alexandre Poletti, Juergen Jenne, Thomas Zilli, Rares Salomir

**Affiliations:** 1Image Guided Interventions Laboratory, GR-949, Faculty of Medicine, University of Geneva, 1205 Geneva, Switzerland; 2Fraunhofer Institute for Digital Medicine MEVIS, Max-von-Laue Str. 2, 28359 Bremen, Germany; 3Division of Radiation Oncology, Avenue de la Roseraie 53, University Hospitals of Geneva, Faculty of Medicine, University of Geneva, 1205 Geneva, Switzerland; 4Division of Radiology, Rue Gabrielle-Perret-Gentil 4, University Hospitals of Geneva, 1205 Geneva, Switzerland; 5Radiation Oncology, Oncology Institute of Southern Switzerland, EOC, 6500 Bellinzona, Switzerland

**Keywords:** hyperthermia, modelling, transperineal, MRI, focused ultrasound, prostate

## Abstract

**Simple Summary:**

Locoregional hyperthermia has been widely used in combination with radiation therapy in several cancer types, including radiorecurrent tumors. Radiorecurrent prostate cancer is a challenging clinical situation, with few treatments available. Reirradiation combined with hyperthermia represents an appealing treatment strategy, although few studies have been reported to date. We aimed to design a specific transperineal focused ultrasound transducer for prostate cancer regional deep hyperthermia treatment, apt to ensure optimal target coverage and patient comfort. The existing prostate transducers (endorectal or transurethral) are not designed for treatments longer than 1h and are able to act upon limited volumes only.

**Abstract:**

Design, embodiment, and experimental study of a novel concept of extracorporeal phased array ultrasound transducer for prostate cancer regional deep hyperthermia treatments using a transperineal acoustic window is presented. An optimized design of hyperthermia applicator was derived from a modelling software where acoustic and thermal fields were computed based on anatomical data. Performance tests have been experimentally conducted on gel phantoms and tissues, under 3T MRI guidance using PRFS thermometry. Feedback controlled hyperthermia (ΔT = 5 °C during 20min) was performed on two ex vivo lamb carcasses with prostate mimicking pelvic tissue, to demonstrate capability of spatio-temporal temperature control and to assess potential risks and side effects. Our optimization approach yielded a therapeutic ultrasound transducer consisting of 192 elements of variable shape and surface, pseudo randomly distributed on 6 columns, using a frequency of 700 kHz. Radius of curvature was 140 mm and active water circulation was included for cooling. The measured focusing capabilities covered a volume of 24 × 50 × 60 mm^3^. Acoustic coupling of excellent quality was achieved. No interference was detected between sonication and MR acquisitions. On ex vivo experiments the target temperature elevation of 5 °C was reached after 5 min and maintained during another 15 min with the predictive temperature controller showing 0.2 °C accuracy. No significant temperature rise was observed on skin and bonny structures. Reported results represent a promising step toward the implementation of transperineal ultrasound hyperthermia in a pilot study of reirradiation in prostate cancer patients.

## 1. Introduction

Prostate cancer is the second most diagnosed cancer among males worldwide and represents more than 1.4 million new cases of cancer in 2020 [1]. Radiotherapy is one of the cornerstone treatments for patients with localized prostate cancer [2]. However, local recurrence occurs in up to 30% of the patients, the greater probability being in patients with high-risk disease pattern [3]. At present, most patients with local recurrence receive long-lasting androgen deprivation therapy, with serious adverse effects such as cardiovascular events and a negative impact on quality of life [4]. However, patients diagnosed with isolated local recurrence can be considered for local salvage treatments, including radical prostatectomy, brachytherapy, high-intensity focused ultrasound (HIFU) or stereotactic body radiotherapy (SBRT) [5,6,7,8,9,10]. The major challenge in treating radiorecurrent prostate cancer is the significant risk of genitourinary (GU) and gastrointestinal (GI) toxicity. The application of ablative dose on the area of tumor recurrence is limited by the dose previously received by organs at risk (rectum, bladder). As no large prospective trial have been performed in this patient population randomizing the different treatment option, a recent meta-analysis showed that SBRT was associated with the lowest rates of both severe GU and GI toxicity (5.6% and 0%, respectively), raising the interest of this approach in a selected population of patients [11]. While a critical need to improve the efficacy of salvage therapy remains, a recent systematic review highlighted the value of combined hyperthermia and radiotherapy in the management of prostate cancer patients [12]. Although limitations exist due to both heterogeneity of trials and time of publication, the combination of radiotherapy and hyperthermia in the definitive or salvage setting has been associated with promising oncological outcomes and no increased toxicity.

Combined radiotherapy and hyperthermia therapy relies on a strong biological basis [13,14] as it demonstrated a complementary action with regard to cell killing by increasing cell radiosensitivity, improving reoxygenation [15] and interfering with crucial DNA repair [16,17,18,19]. Improvement of tumor oxygenation after mild hyperthermia, which lasts for as long as 24-48h after heating, may increase the effects of radiation therapy [20]. Kok et al. [21] estimated that hyperthermia could provide an equivalent delivered dose of 10Gy higher than radiotherapy alone, using the linear-quadratic (LQ) equation. Yet, the clinical implementation of combined radiotherapy and hyperthermia remains scarce worldwide, due to several technical challenges.

Commercial devices are available for the induction of deep and superficial hyperthermia; however, using microwaves for heating tumor tissues with uniformity and precision as well as for measuring temperature and thermal dose remains nontrivial. Loco-regional heating devices are usually composed of several radiofrequency antennas organized in multiple rings [22,23,24]. Using these, the creation of unwanted hot spots in normal bystander tissue is often inevitable [25]. Unlike commercial devices delivering deep regional hyperthermia using microwaves or radiofrequency our device delivers thermal energy in deep tissues by highly focused ultrasound. Focused ultrasound is the only known technology that enables sharply localized energy deposition inside deep tissues in a non-invasive manner [26,27,28]. Magnetic resonance-guided focused ultrasound technology (MRgFUS) employing modern phased-array transducers, has several fundamental advantages: accurate delineation and the possibility to custom shape the zone to be heated, real-time temperature monitoring (multi-planar or 3D) and closed loop automatic feedback to survey the delivery of the prescribed hyperthermia plan [29]. 

Ultrasound based thermal therapy of the prostate is currently performed by two techniques: transrectal and trans-urethral focused beam application. Both have shown promising results over the past years in more than 100 sites across the world [30,31]. Several commercially therapy system exist such as the FocalOne or Ablatherm (EDAP-Technomed, Lyon, France), Sonablate HIFU (Sonablate, Charlotte, NC, USA), with follow-up studies 2–5 years post HIFU-treatments indicating stable low levels of PSA and significantly increased survival rate in patients [32,33]. MR-guided therapeutic ultrasound was suggested to deliver prostate hyperthermia via the endorectal route [34], using a commercial MR-guided endorectal HIFU ablation array. Numerical simulation and non-perfused phantom experiments suggested that the device could be controlled in principle for delivering continuous hyperthermia in prostate, while working within operational constraints. MR-guided HIFU based hyperthermia was further investigated in a theoretical study without experimental output on the targetability of late-stage cervical cancer by with another commercial device, FDA/CE approved for ablation of uterine fibroids [35].

The aforementioned devices are specifically dedicated to thermal ablation; that is intended for short treatment durations at high powers. On the other side, moderate hyperthermia entails prolonged treatment duration of about 1 hour, the complexity of heating a large volume homogeneously and requires optimal patient comfort to sustain the procedure. No study to date has used an external ultrasound transducer to induce prostate hyperthermia, with the aim to improve targeting and patient comfort.

Performing transperineal hyperthermia on the prostate is a challenge. As the target is away from the transducer the main risks are the pelvic bones and the entry window becoming overheated. Another factor is the difficulty in maintaining a stable temperature due to the increased perfusion of the tumour tissue when the temperature rises [36,37].

This article presents a novel design of ultrasound based, MR compatible extracorporeal system, aiming to deliver in an adjuvant setting mild local hyperthermia to radiotherapy for prostate and eventual pelvic application, while minimizing the risks for adverse effects in this configuration.

The first step was to design a transperineal focused ultrasound transducer to induce prostate hyperthermia throughout the pelvic window and a software was developed to allow for the computation of acoustic and thermal fields based on anatomical data, considering two models of tissue perfusion. The transducer was designed to provide prostate hyperthermia in application that may exceed 60 min. The second step was to determine the performance of the transducer and to ensure technical feasibility easiness of handling in a clinical context. The last step consisted in inducing ex vivo closed loop hyperthermia treatment on lamb carcasses with a tissue mimicking prostate.

## 2. Materials and Methods

### 2.1. Transducer Design

#### 2.1.1. Features

The transducer was designed by the authors and manufactured by Imasonic company (Besançon, France) to fit in the pelvic window when the patient is lay on the back. The device is adjusted to create mild hyperthermia of 4–6 °C above the baseline tissue temperature, during 30 to over 60 min, to sensitize tumor cells before radiation therapy. Various constraints linked to the treatments had to be taken into consideration in our model, as patients would be treated within the magnet of a high field closed bore MRI scanner, including anatomic constraints for the ultrasound beam propagation like surrounding bony structures, when designing the software used to determine transducer parameters. Three thermocouples were added on the active surface to ensure a standard temperature elevation, one thermocouple in the midsection of the active surface, the other two at the edges of the surface (bottom right and top left). A cooling system was designed, by adding a biocompatible membrane on the front side of the transducer, to ensure ultrasound propagation and convective cooling of the transducer surface.

#### 2.1.2. Images Segmentation

In a first step, using computed tomography (CT) images of twelve anonymized patients (FOV = 512 mm, pixel spacing = 0.97 mm, filter = flat, slice thickness = 2) from a retrospective database, different pelvic structures and organs (bone, muscle, other soft tissues) were identified and a theoretical acoustic window determined. Then, MRI images of two volunteers were prospectively acquired and used to provide a more realistic numerical model. Written informed consent was obtained prior to this acquisition. As both bone and air produce weak signals on T1-weighted turbo spin echo MR sequences (repetition time (TR) = 783 ms, echo time (TE) = 12 ms, bandwidth (BW) =170 Hz/pixel, acquired resolution= 1.56 mm × 1.17 mm × 3 mm, FOV = 512 mm, acquisition time = 185s), segmentation of these materials showed to be particularly challenging, especially when located in close proximity one to the other. A target region was identified on the images, after hip, pelvis and coccyx segmentation performed by a clinician. The skin was then manually segment using the SAFIR (Software Assistant for Interventional Radiology, Fraunhofer MEVIS, Bremen, Germany). The segmentations provide curves which delineate different anatomical structures. From these curves, a region-growing algorithm converted 2D data into labelled volumetric data.

#### 2.1.3. Acoustic and Thermal Fields Modelling

A modelling software, where acoustic and thermal fields were computed based on anatomical data, was created to design a new transperineal therapeutic ultrasound transducer. 

At the intensities and durations considered propagation should be modelled as linear and continuous wave which can handle inhomogeneous materials. The linear acoustic wave equation with a general frequency-dependent absorption law takes the form
(1)$$\frac{\partial^{2}p}{\partial t^{2}}=c\nabla^{2}p+L\left[p\right]$$
where *c* = *c*(x,y,z) is the spatially varying speed of sound (m/s) and L is an attenuation operator which, for a harmonic wave with an angular frequency ω = 2πf (rad/s), satisfies for $\alpha \left(\omega \right)=\alpha_{0}\left(x,y,z\right)\omega^{\eta \left(x,y,z\right)}$
(2)$$\left|p\left(x,\;y,\;z+\delta z\right)\right|=\left|p\left(x,\;y,\;z\right)\right|e^{-\alpha \left(\omega \right)\delta z}$$
where $\alpha_{0}$ is the attenuation coefficient [(rad/s)^-η^ Np/m] and η is the (dimensionless) attenuation exponent.

Given the large volumes treated in hyperthermia a computationally efficient scheme is required, furthermore, the scheme must account for heterogeneities in soft tissue. Considering all factors, the hybrid angular spectrum [38] was employed as it satisfied all criteria. 

In the hybrid angular spectrum an input plane is defined and the forward propagation of the acoustic field is computed. Inhomogeneity results in a hybrid method: one part of the simulation, accounting for attenuation and propagation through an averaged material, is performed in the spatial domain, whereas another part of the simulation which accounts for propagation through inhomogeneous material, is performed in the Fourier domain. For more details, refer to the Appendix A.

On segmented Digital imaging and communications in medicine (DICOM) MR images, the center of each voxel defines a grid point. Each voxel of each segmented DICOM image has a material label, with associated acoustic and thermal properties [39], see Table 1.

The transducer is positioned in such a way as to reduce exposure to the pelvis. Simulations provided a maximal time averaged-acoustic intensity on the surface of the bones. Thus, it is justified to neglect any shear-wave generation, as well as scatter from the pelvis. 

When the phase of the pressure field and the particle velocity are in phase, the heat source can be correctly characterized by the plane wave assumption. The source term for the computation of the thermal field is given by
(3)$$q\left(x,y,z,t\right)=\frac{\alpha \left(x,y,z,\right)}{\rho \left(x,y,z\right)\;c\left(x,y,z\right)}{\left|p\;\left(x,y,z,\;t\right)\right|}^{2}$$

The size of the computation domain is (256, 256, 256), so that the spatial resolution is (0.781, 0.781, 1.024) mm, so that the physical size of the computational domain is (20, 20, 30) cm. At the operating frequency of the transducer, this indicates that the lowest point-per-wavelength is 2.93. (cf. Table 1), sufficient to accurately resolve the computation of the entire acoustic field.

The multi-element phased-array transducer is characterized using the plane piston assumption. Each transducer element is discretized into smaller computational elements, whose phase and magnitude are computed using the Rayleigh integral approximation of the Kirchhoff integral [41].

The geometric center of the transducer defines the center of the computational domain, and the beam axis of the transducer defines the z axis in the computational domain. When the position and orientation of the transducer are modified, the computational domain is recomputed using a nearest neighbor interpolation to assign material labels to the new grid points. 

A convergence analysis was performed to show that the size of the computation domain is sufficiently large to fully capture the acoustic field. Additionally, a convergence analysis was performed by refining the computational domain, while maintaining the labels for the grid points, in order to show that the acoustic field was unaffected by increasing the spatial resolution.

The thermal field was computed using the Pennes bioheat transfer equation [42].
(4)$$c_{v}\rho \;\frac{\partial T\left(x,y,z,t\right)}{\partial t}=\nabla \cdot \left(\kappa \left(x,y,z\right)\nabla T\left(x,y,z,t\right)\right)-\nu \left(T\left(x,y,z,t\right)-T_{0}\right)+q\left(x,y,z,t\right)$$
where $c_{v}$ is the specific heat capacity of tissue [J/(kg K)], ρ is the mass density [kg/ m^3^ ], *κ* is the thermal conductivity of tissue [W/(m K)], *ν* is the isotropic bulk perfusion parameter which accounts for energy loss [J/(K m^3^ s)], $T_{0}$= 37 °C is the equilibrium body temperature, and q [W/m^3^] is the heat source term from Equation (3). Note that $\nu =c_{v}^{b}\rho^{b}\bar{\nu }$, where $c_{v}^{b}$ and $\rho^{b}$ are the specific heat capacity and thermal conductivity of blood, and $\bar{\nu }$ is the perfusion rate per unit volume of tissue [1/s]. 

The numerical computation has explicitly considered the tissue perfusion as written in Equation (4). Two cases were considered: (i) a uniform value of perfusion set to 0.0011 s^-1^ according to Table 1, and (ii) a temperature-dependent perfusion as a quadratic model [36] where the interval of perfusion values was set from 0.0011 s^−1^ to 0.0083 s^−1^. The highest value was taken from Vulpen and al. [37] and the quadratic function, detailed in Section 3.1, was evaluated for each voxel with the temperature value from the previous time step of the iterative calculation.

The bioheat equation is computed using an explicit finite-difference time-domain scheme, considering the spatial variations of the thermal properties due to differing tissue types. The first- and second-order spatial derivatives are approximated using first-order accurate backwards and second-order accurate central differencing methods, respectively. The temporal derivative is approximated using a forward differencing method. Dirichlet boundary conditions are applied, with the body temperature, consistent with the acoustic field. The characteristic length-scales associated with the heat equation are by far larger than those of the acoustic equation. However, to preserve the heterogeneous structure of the material properties, thermal simulations are performed on the same grid as the acoustic simulations, rather than being performed on a coarser grid. While this preserves the anatomical accuracy of the simulations, through the Courant-Friedrichs-Lewy stability condition, it results in a temporal resolution being finer than it would be on a coarser grid. Given the duration of the exposures, this results in an increase in computational time. However, this proved not to be a significant problem in our model, as the simulations were performed on a graphical processing unit. The thermal code itself has been extensively validated in the context of cryoablation [43] and radio-frequency ablation.

#### 2.1.4. Numerical Optimization of the HIFU Transducer

After verification of the software using known models, volunteer images were used to determine the size, shape, layout and operating frequency of our transducer. Initial candidate transducers had variable frequency and convexity (defined by radius of curvatures $R_{x}$ and $R_{y}$ in x and y directions, respectively). From segmented DICOM MR images the center of each voxel defined a grid point. Each voxel of the segmented DICOM images was assigned a material label, with associated acoustic and thermal properties [39]. The transducer was positioned in such a way as to reduce thermal exposure to pelvic bones (Figure 1). 

Simulations generated the maximal time averaged-acoustic intensity on the surface of the bones. Thus, it appeared justified to neglect any shear-wave generation, and scatter. The accurate modelisation of acoustic pressures incident on the surface of the bone was of importance given that proton resonance frequency shift thermometry cannot measure temperatures in either cortical bone or bone marrow.

### 2.2. Driving Electronics

The described therapeutic ultrasound device was powered by a high-power beam former (10W/channel, 256 channel beam former, frequency 0.5 to 3 MHz, computer-controlled amplitude, phase and frequency, forward and reflected power monitoring, external triggering) manufactured by Image Guided Therapy (Pessac, France) using a proprietary software (Thermoguide, version 1.3.7). An impedance machine unit, manufactured by Image Guided Therapy, was positioned outside the 10mT MRI field line and multi coaxial cable length between the impedance machine unit and the transducer was 3.5m.

### 2.3. Software

The HIFU beam former was controlled by dedicated software via an ethernet connection. A multi-thread application was written in Python programming language (Python Software, version 3.7, Beaverton, OR, USA) for Windows 7 and 10, to enable the control of the power generator output in real time, using the manufacturer’s drivers (“IGTFUS” library). The program allows for a selection of the transducer that will be used, and also to easily switch between transducers. The main window is an overview of the transducer, which permits to initiate different shooting modes, view graphically if an element is disabled, disable a channel, and finally to set shooting parameters such as power and x-y-z steering. In continuous shooting mode, the graphs allow to visualize the progress of temperature and applied power in real time. The program records all parameters applied during a treatment session, allowing for post-treatment analyses of applied powers and settings.

### 2.4. Positioning Device of the Applicator

The transducer screws onto a holder which can be stepwise rotated in the medial-lateral direction to compensate for limited electronic steering of the focused ultrasound beam on the short axis of the applicator. The height of the applicator is set with spacers such that it aligns to the patient’s pelvic window. Electronic steering along the ventral dorsal axis, corresponding to the applicator’s long axis, allows for proper targeting of the tumor. The transducer and holder are inserted with 4 sliders into a base frame that is attached to the MRI table. The sliders are attached with elastic bands fixed to the base frame to provide pressure in the cranio-caudal direction for proper coupling of the ultrasound beam with body tissue.

### 2.5. MRI Guidance of Local Hyperthermia

During experiments, performed under 3T MR guidance (Prisma Fit, Siemens, Erlangen, Germany), the geometry of therapeutic ultrasound transducer and used tissue sample was verified on a highly resolved T1-weighted 3D gradient echo (GRE) sequence (TE = 3 ms, TR = 6.88 ms, flip angle = 10°, 0.8 mm isotropic voxel size, acquisition time 120 s), and data was acquired with a combination of a standard spine coil and a flexible body coil. The temperature elevation (ΔT) in tissue mimicking gel was calculated using the proton resonance frequency shift (PRFS) method, based on the temperature-related changes in the phase of the gradient echo signal (three orthogonal slices acquired interleaved, segmented echo-planar-imaging (EPI) factor = 7, temporal resolution 2.7 s, FOV = 128 × 128 mm^2^, slice thickness = 4 mm, in plane resolution 1 mm, TR = 50 ms, TE = 10 ms, flip angle = 20, BW = 164 Hz, spectral-selective fat suppression).

### 2.6. Performance Test

Transducer heating and quality control of focusing were assessed with the transducer immersed in a degassed still cold-water bath. Sonications were performed on tissue mimicking gels composed of 2% *v*/*v* agar, 8% *v*/*v* glycerol and 12% *v*/*v* powder milk [44]. 

The expected position of the focal point without lateral steering was determined from the 3D T1-weighted sequence, using the orthogonal symmetry planes of the transducer. The intersection of these planes defined the main acoustic axis. A third plane, perpendicular to the two others was added at the level of the focal point. These three reciprocally orthogonal planes also defined the geometry of the MR thermometry slices.

Several sonications were performed to control the steering and the precision of the focal point, 9 sonications with steering along the x axis, 11 sonication with steering along the y axis and 5 sonication with steering along the z axis. When steering was used, the three MR thermometry slices were shifted accordingly. Then, three sonication of 50 W were performed at the focal point (0,0,90) during 10 min. A sonication at 250 W was also performed for 30s in order to test and register the temperature measured by the three thermocouples embedded in the transducer. 

A predictive temperature controller was used [45], which automatically adjusted the acoustic energy deposition during a long sonication (30 min) and the delivered acoustic power ranged between 30 and 50W. The controlled variable was the MR temperature averaged in a region of interest (ROI) of at least four pixels around the selected focal point. A baseline correction of magnetic field drift was applied, using a large ROI outside the focused ultrasound beam pathway. Offline analysis of thermal maps was performed using Matlab R2021a (MathWorks, Natick, MA, USA).

### 2.7. Ex-Vivo Studies

Ex vivo tests were conducted to verify the performance of the transducer and ensure its safe use, especially regarding the absence of secondary heating of pelvic bones. 

An anatomic model mimicking the pelvic window and the prostate bed was used for this purpose, consisting of a lamb carcass (obtained from a butchery) with a pelvic cavity filled with standard ultrasonic gel and turkey meat mimicking the acoustic capacity of the prostate. The fresh lamb carcasses (*N* = 2) were laid on the back on the MRI table, with the hind legs approximated to the abdomen. This position ensured a larger pelvic window, allowing for better access of the focused ultrasound transducer. Following that, the pelvic cavity was filled with acoustic gel and then the turkey breast was inserted into this cavity serving as a tissue model of prostate or prostate bed (Figure 2). Acoustic coupling between the therapeutic transducer and the skin was achieved with the biocompatible membrane inserted on the front side of the transducer and filled with cold degassed water. Standard ultrasonic gel was applied between the front side membrane and the tissue.

Targeting was achieved with the high resolution T1-weighted 3D GRE sequence (described in Section 2.5) (Figure 2) inside the same 3T clinical scanner used for the performance tests, using a receive flexible body coil. By means of a transducer holder, the pelvic window was aligned as close as possible with the focused ultrasound beam axis. The focal point was positioned 90 mm from the skin where cancer recurrence following prostatectomy most frequently occurs.

Standard PRFS thermometry using the segmented GRE-EPI sequence (described in Section 2.5) with lipid signal suppression and $B_{0}$-drift compensation was performed parallelly and perpendicularly to the target. The focal point position was confirmed using a pilot sonication of 50W for 30 second. Prescribed hyperthermia was defined as a uniform 5°C temperature elevation within the tissue mimicking prostate for 15 minutes. In the first condition the HIFU targeted a fixed focal point and in the second condition, the energy was delivered in a circular pattern defined by a 4mm diameter circle described by 16 points. A predictive temperature controller was implemented, automatically adjusting the ultrasound energy deposition. The temperature controller considered the predicted asymptotic level of temperature elevation, as determined from a sliding temporal window of observation. Full details are provided by Guillemin et al. [45].

## 3. Results

### 3.1. Transducer Design

The following parameters were achieved through our optimization method to create a transducer suited to treat recurring prostate cancer. The transducer is based on a cylindrical surface with a concave geometry, creating a natural focus along a 140 mm left-to-right dimension, remaining relatively flat with 60 mm in the anterior-posterior dimension (Figure 3a).

With the numerical model, the operating frequency was chosen at 700 kHz, especially to limit heating of pelvic bones. The phased-array transducer has 192 elements, pseudo-randomly distributed on 6 columns of 29/31/34/34/31/29 elements (see Figure 3c). The average active surface of one element is 8.70mm² (6.3–11.0 mm²) for a total of 192 elements covering 1670 mm² of active surface. The amplitude of radio-frequency (RF) excitation of each active element was weighted by the square root of the element’s surface, to create a uniform acoustic source. The cooling system’s flow rate was 1.2 L/min, and the water temperature 20 °C (Figure 3b).

According to the numerical calculations, under realistic perfused conditions, the device is able to deliver a temperature elevation of +6 °C when a conservative value of perfusion is set according to Table 1 and, respectively, +5.7 °C when a non-linear extreme heat sink is considered according to Figure 4a. In both cases, owning to the closed loop temperature controller (Figure 4c,d), a localized hyperthermia at 90 mm depth inside the tissue is achieved (see Figure 4b). 

There is no model-based evidence of overheating the entrance window, owning to the active cooling of the body surface by the circulating water, set here at 20 °C, in front of the applicator. Of interest, given the long duration of the sonication, the superficial cooling becomes effective also for deeper layers of tissue, see Figure 4b, since a steady state temperature gradient is being established. While the maximum temperature elevation is comparable, the geometry of the thermal buildup is significantly modified by the tissue perfusion, that yields a narrower longitudinal profile and the estimated energy emission is doubled.

### 3.2. Performance Test in Tissue Mimicking Gel

The 192 elements were individually electronically adaptable to set a focal point in a volume of 24 × 50 × 60 mm at -3dB around the natural focal point. The offset between the prescribed position and the focal point was determined in all three axes being on the order of one pixel (Figure 5).

The temperature elevation of the material measured by the embedded thermocouples on three points of the transducer was 4.1 °C, 4.7 °C and 3.8 °C, respectively, at 250 W applied during 30 s. This range is definitely at no risk in terms of thermal stress. 

PRFS thermometry performed in the 3T MR scanner on tissue mimicking gels demonstrated no interferences and no artefacts on MR images during concurrent sonication of up to 200 W. Analysis of temperature maps confirmed the absence of significant thermal effects from secondary lobes.

Pixel-wise temporal standard deviation of MR thermometry in the tissue mimicking gel was on average 0.2 °C. A steady-state regimen during 15 min long hyperthermia, defined by an absolute offset of less than 0.2 °C between the actual temperature elevation and the predefined target (5 °C), was obtained in 260 s in average. 

### 3.3. Ex Vivo Test

Experiments on lamb carcasses succeeded to demonstrate the technical feasibility of our protocol, showing effective targeting of a tissue mimicking prostate for long sonication, and the absence of interferences between the MR scanner and therapeutic transducer. 

The optimal focal point positioning was determined by the acoustic window and the depth of the target (tissue mimicking prostate), here it was at 90 mm for the lamb 1 and at 85 mm for the lamb 2.

The average steady-state temperature elevation for a fixed focal point and 15 min long hyperthermia, was 5.06 ± 23 °C for lamb 1 and for a circular pattern of foci was 5.09 ± 0.29 °C for lamb 2 (Figure 6). 

Our predictive controller suppressed fluctuations of temperature in ex vivo tissue and demonstrated typical accuracy of 0.2 °C, clearly sufficient for clinical application of the device.

PRFS thermometry data obtained in soft tissue voxels adjacent to the bones preserved sufficient MR signal to determine the estimated temperature elevation on the pelvis (coxal) bones themselves. The corresponding measured temperature elevation from the ex vivo model was found to be in the anticipated range by the numerical model in the human anatomy (mean 0.52 °C, min −1.1 °C, max 0.9 °C). This increase is not significant and allowed us to conclude that no overheating of the pelvic bones occurred.

## 4. Discussion

To our knowledge, this is the first study to investigate therapeutic ultrasound application with a transperineal approach, for prostate or pelvic cancer. Up to this date, only phantom studies have been conducted for the induction of hyperthermia, mimicking the treatment of prostate cancer, using endorectal phased-array systems [46], originally designed specifically for thermal ablation where sonication durations are short, and the power used is higher than that typically used in a radiosensitizing hyperthermia session. To achieve precision, focal points of ablation dedicated HIFU transducers are very small, and to treat larger regions, dozens or even hundreds of iterations are necessary [47]. Contrarily, for a radiosensitizing hyperthermia treatment, the sharpness of the thermal build-up is less important, as the aim is to achieve simultaneous and uniform heating of a volume over a prolonged period of time. To the best of our knowledge, the system presented here is the first one specifically created for such prolonged sonications that may exceed 1h, but also in the overall objective to improve the patient comfort.

Overall, the design of our MRI-associated therapeutic ultrasound applicator took into consideration various factors resulting in an optimized compromise between acoustic requirements, suitable ultrasound production technology, pelvic anatomy and patient comfort. The convex shape of the outer membrane was chosen to ensure an anatomically correct coupling of the device to perineal skin [48]. The water circulation system prevents the accumulation of air bubbles in front of the active part of the transducer. The thermocouples were embedded in the piezo-composite material to ensure that heating of the elements themselves remains low and consequently also the risk of skin burns to the patient. Moreover, information from embedded thermocouples outside the sonication was indicative of the water temperature of the cooling system, enabling adjustment of the temperature if needed. As the tissue temperature rises induced by the acoustic field are small and lasts for long duration, the thermal field remains sensitive to changes in surrounding temperature fields. Thus, ensuring that the skin temperature is kept at physiological levels helps preventing unwanted hyperthermia of near field tissue.

Both lamb carcass experiments confirmed the accuracy of the sonication. The temperature elevation was maintained at 5 °C with better than 0.3 °C accuracy during 15 min. It could be concluded that when delivering a low acoustic power over a prolonged duration, heat conduction produces a quasi-stationary thermal build-up. Moreover, our experiments validated the safety of the technique as no significant deposition of heat at the pelvic bones’ surface was detected, neither during the experiments with the PRFS method, nor after the experiments by direct visual observation.

This system was also designed to fit with a personalized 3D-printed pelvic immobilization device created for combined radiotherapy and MRgFUS deep HT treatments in the pelvic region [49]. The device allowed for adaption to an individual patient’s anatomy with millimeter accuracy and ensure patient safety during the entire duration of a treatment. Moreover, the immobilization device permitted a fast and precise positioning of patients and an optimal inter- and intra-fractional control of motion during hyperthermia treatments, including modern adaptive radiotherapy techniques with MR-LINACs [50]. 

At the pressures and exposure durations considered for radiosensitizing hyperthermia a linear, continuous wave is well-suited to accurately model the acoustic field. As the transducer is convex and the near field acoustic energy is distributed over a wide area and as the transducer is coupled to the patient through water, it is assumed that no significant defocusing occurs at interfaces.

Prostate cancer is not usually characterized by distinct solid tumors, and the cancer regions can have heterogeneous physical properties and blood flow [51]. The models developed in this study do not consider heterogeneity of physiological parameters which can impact the quality of a hyperthermia treatment. However, this technology is used under MRI in real time, and the feedback control, allowing the hyperthermia induction system to adapt to physiological changes. 

Online PRFS MR thermometry needs to be performed continuously during hyperthermia using gradient spoiled GRE sequence, which in general has a low specific absorption rate (SAR). The required sampling rate of MR thermometry acquisition in the context of mild hyperthermia is not critical, further lowering the SAR. Overall, additional RF heating by the MRI on the top of ultrasound induced hyperthermia is not considered a significant issue.

One of the limitations of our device is that the model assumes one-way propagation of ultrasound waved, and that reflections from scattering are not considered. This may lead to some slight under-estimation of focal intensities. Additionally, while the transducer is placed in such a way as to minimize the acoustic beam targeting of bones, calcifications [52] could possibly affect adversely temperature profiles. Given the ex vivo targeted tissue with only 5 to 6 °C temperature elevation above the room temperature, no histological studies were considered in this work. 

While this pilot study represents a proof of concept, several points must be taken into consideration before clinical application. Experiments on perfused tissue models, described for instance by Lorton et al. [53], and ultimately in vivo studies should be further performed in order to demonstrate the capacity of the transducer to deliver accurate and safe hyperthermia under more challenging conditions. Moreover, we need to analyze and measure the accuracy of pelvic PRFS MR thermometry in humans using the full setup, with the transducer in its holder positioned likewise to a real treatment condition. 

Locally recurrent prostate cancer after a primary radiotherapy raises significant challenges for salvage reirradiation and may benefit from the proposed approach. However, possible application of our combined treatment may include definitive radiotherapy for patients with locally advanced tumors, salvage radiotherapy for macroscopic relapse in the prostate bed after radical prostatectomy, or locally advanced pelvic mass, if acoustically accessible. 

Although retreatment with SBRT techniques has been shown to provide encouraging results in terms of biochemical control, the risk of subsequent local failures and treatment related toxicity is not negligible [9,11,53]. As dose escalation is required even in the reirradiation salvage setting, the combination of radiotherapy with hyperthermia may help improving the therapeutic ratio of these patients. Finally, our approach can synergistically work together with other techniques able to reduce the potential radiation-induced toxicities as the injection of hydrogel spacers between prostate and rectum [54,55], that will also reduce the risk of ultrasound heat deposition at such tissue-gas interfaces.

## 5. Conclusions

In conclusion, in this study we developed a new concept of a transperineal focused ultrasound transducer for the induction of regional deep hyperthermia treatments of radiorecurrent prostate cancer. Using MR thermometry, we demonstrated that the phased array transducer allowed a precise delivery of hyperthermia inside the target volume, ensuring a fast patient setup by means of a transducer holder. Validation of the results in live perfused tissue is needed before routine use in clinical practice can be envisaged.

## Figures and Tables

**Figure 1 cancers-15-00163-f001:**
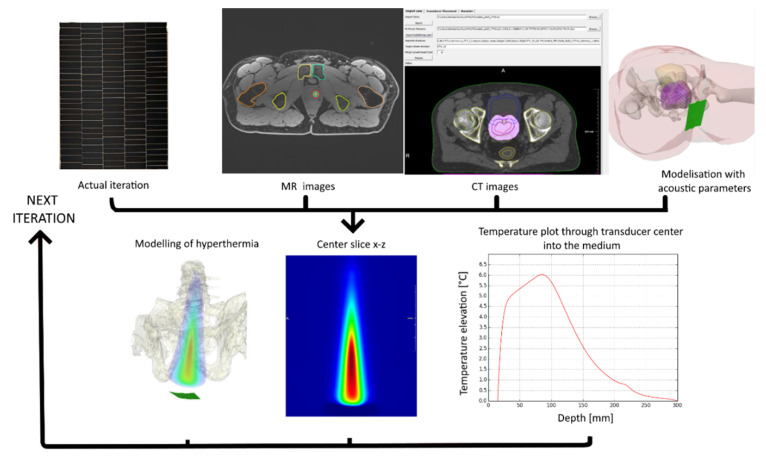
Iterative process of transducer design. A first design was tested with images from a CT scan of a patient and a modelisation software. The obtained results permitted to adjust individual elements’ position on the transducer.

**Figure 2 cancers-15-00163-f002:**
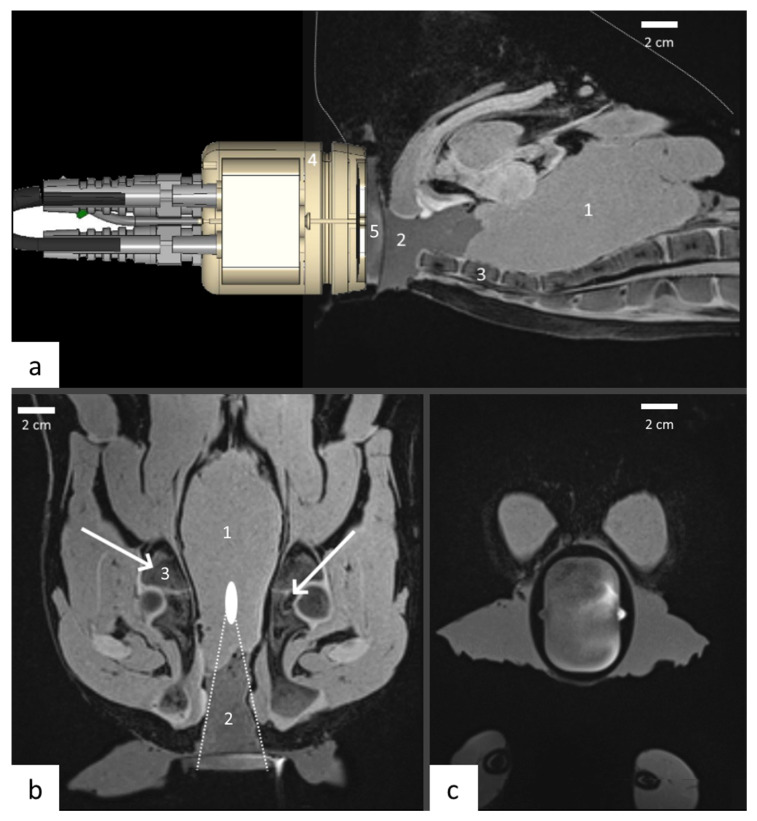
Positioning of the phased-array transducer on carcass lamb: (**a**) sagittal, (**b**) coronal and (**c**) axial sections in the T1w 3D GRE data. Transducer (4) was placed in the axis of the simulated target (1), the coupling between the target and the transducer was realized with standard ultrasonic gel (2) and the water flow compartment in front of the applicator (5). A special attention was paid to thermal energy deposition on bone (3) that is indicated with white arrows.

**Figure 3 cancers-15-00163-f003:**
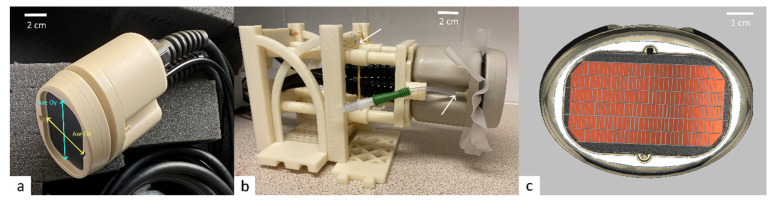
The novel hyperthermia transducer for prostate cancer. (**a**) Transperineal phased array transducer; (**b**) customized 3D printed holder and water circuit (white arrows); (**c**) padding of the active surface with acoustic elements.

**Figure 4 cancers-15-00163-f004:**
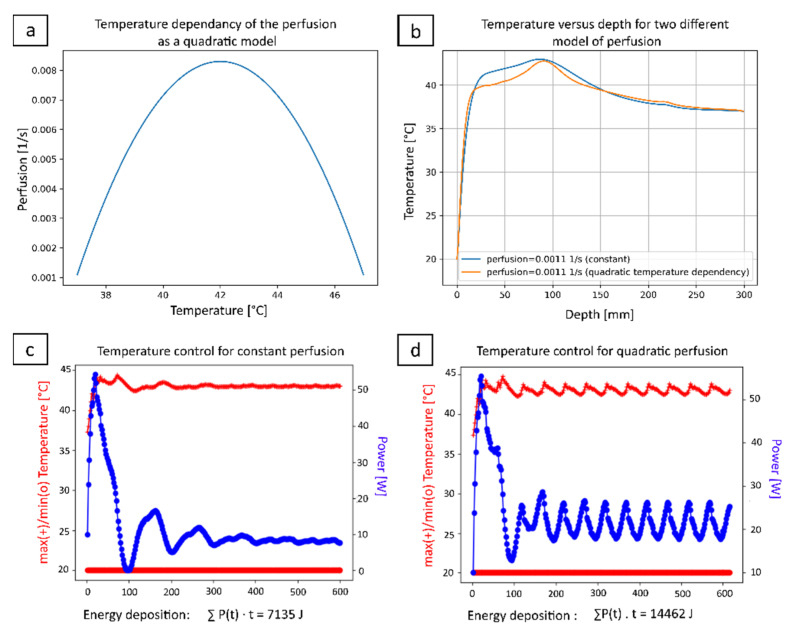
Prediction of the performance of the device in perfused tissue, for a constant perfusion model *p* = 0.0011 $s^{-1}$ and, respectively, for a quadratic temperature-dependant perfusion model, shown in (**a**). (**b**) Simulation of the temperature profile along the acoustic axis during hyperthermia for the two perfusion models; (**c**,**d**) Plot of the estimated action of the feedback controller for both cases, with cumulated energy emission reaching 7.1 kJ and 14.5 kJ, respectively.

**Figure 5 cancers-15-00163-f005:**
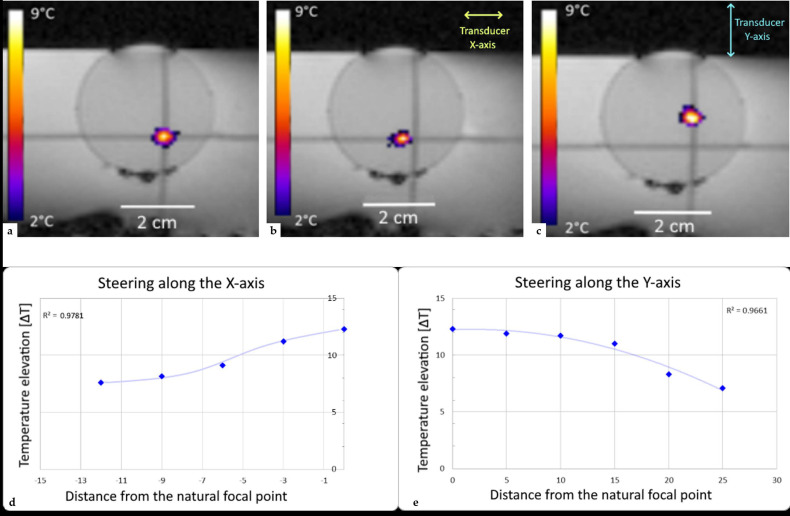
Study of steering capacities in the X and Y axes. (**a**) Sonication at the symmetry axis (**b**) sonication with steering of 8mm in the X-axis, and (**c**) sonication with steering of 15 mm in the Y-axis. (**d**,**e**) Tables of steering capacities.

**Figure 6 cancers-15-00163-f006:**
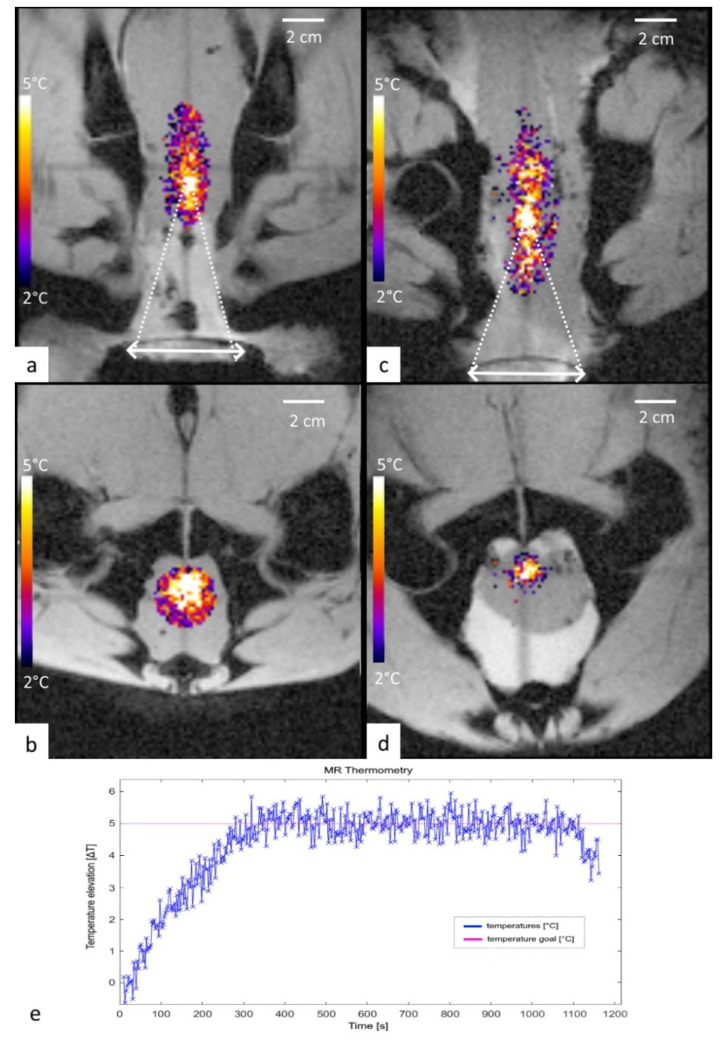
Experiment results for both carcass lamb. (**a**,**b**) MR thermometry in the coronal and axial geometry, a single focal point sonication shown at 6 min time point for lamb 1. (**c**,**d**) MR thermometry in the coronal and axial geometry, a circular pattern of foci shown at 6 min time point for lamb 2. (**e**) Temperature versus time plot for lamb 1 at the focal point.

**Table 1 cancers-15-00163-t001:** Acoustic and thermal properties for tissue types. Note that the value of absorption indicated is based on an operating frequency of 0.7 MHz. ρ is the mass density, c is the speed of sound, α is the ultrasound attenuation, $c_{v}$ is the specific heat capacity of tissue, κ is the thermal conductivity of tissue, ν is the isotropic bulk perfusion parameter which accounts for energy loss and $\bar{\nu }$ is the perfusion rate per unit volume of tissue. The $\bar{\nu }$ for soft tissue was taken from Tilly et al. [40].

Tissue	ρ [kg/m^3^]	c [m/s]	α [Np/m]	$c_{v}\;[J/\;(\mathrm{kg}\;\cdot \;K)]$	κ [W/ (m · K)]	$\bar{\nu }$ $\left[s^{-1}\right]$
Water	1000	1050	0.006	4180	0.621	-
Soft Tissue	1087	1577	6.00	3668	0.475	0.0011
Bone	1900	2990	60.0	1370	0.0435	0

## Data Availability

MR thermometry data will be provided on reasonable request.

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
