# Peer review of "A Novel Concept of Transperineal Focused Ultrasound Transducer for Prostate Cancer Local Deep Hyperthermia Treatments"

_cancers, 2022, doi:10.3390/cancers15010163_

Round 1

Reviewer 1 Report

The authors present a new technical development to be able to induce moderate hyperthermia in larger volumes and  for longer times (~1h) not possible using existing devices that are dedicated for thermal ablation using ultrasound. The manuscript describes the technical and software development, simulations and valuations of the heating procedure. The manuscript is clear and well written and I have really no comments for improvement except two minor ones. I recommend publication of the work presented. 

1. Please provide a legend to table 1

2. I agree that the effect of RF heating by the MRI during the experimental conditions described in the manuscript is not a significant issue, but I like to see that mentioned somehow in the text to remind the readers of that.

Author Response

The authors present a new technical development to be able to induce moderate hyperthermia in larger volumes and for longer times (~1h) not possible using existing devices that are dedicated for thermal ablation using ultrasound. The manuscript describes the technical and software development, simulations, and valuations of the heating procedure. The manuscript is clear and well written, and I have really no comments for improvement except two minor ones. I recommend publication of the work presented. 

Point 1.1: Please provide a legend to table 1

Dear Reviewer, thank you for your comment.

Answer 1.1: We are sorry for this omission; we added a legend to table 1 (L202).

Point 1.2: I agree that the effect of RF heating by the MRI during the experimental conditions described in the manuscript is not a significant issue, but I like to see that mentioned somehow in the text to remind the readers of that.

Answer 1.2: Thank you for pointing this out. MR thermometry was performed continuously during hyperthermia using PRFS gradient spoiled GRE sequence, which in general has a low SAR and the required sampling rate is not very high as the hyperthermia is long. Indeed, this should not significantly modify the heat deposition from the main source (ultrasound) but it worth mentioning, please find new statement (L 524).

Reviewer 2 Report

The manuscript is interesting and well written; the discussion should be improved adding the role of hydrogel spacer injection beetwen prostate and rectum to reduce side effects (Pepe P et al Clinical outcomes of hydrogel spacer injection space oar in men submitted to hyfractionated radiotherapy for prostate cancer. In Vivo 202135: 3385-add in the references)

Author Response

The manuscript is interesting and well written; the discussion should be improved adding the role of hydrogel spacer injection beetwen prostate and rectum to reduce side effects (Pepe P et al Clinical outcomes of hydrogel spacer injection space oar in men submitted to hyfractionated radiotherapy for prostate cancer. In Vivo 202135: 3385-add in the references)

Dear reviewer, thank you for your comment and for the suggestion. After discussed with the radio-oncologist involved in this study and considering the benefit-risk equation for each patient, the injection of hydrogel spacer could be used in this project. In addition to reduce acute and late GU and GI toxicities due to radiotherapy, by spacing the prostate away from the rectum this method can limit the risks of ultrasound propagation in the rectum. We added in the discussion two references on the role hydrogel spacers between prostate and rectum (L 557).

Reviewer 3 Report

The authors investigated the basic performance experiments of the transperineal focused ultrasound transducer. They concluded that these results represent a promising step toward the implementation of transperineal ultrasound hyperthermia in a pilot study of reirradiation in patients with prostate cancer.

This study is a very important and interesting study. I think this paper is very well written.

If the authors performed the experiments on tissues, please add figures of the histological changes. If not, please add that no histological studies have been performed in the limitation section.

Author Response

The authors investigated the basic performance experiments of the transperineal focused ultrasound transducer. They concluded that these results represent a promising step toward the implementation of transperineal ultrasound hyperthermia in a pilot study of reirradiation in patients with prostate cancer.

This study is a very important and interesting study. I think this paper is very well written.

Point 3.1: If the authors performed the experiments on tissues, please add figures of the histological changes. If not, please add that no histological studies have been performed in the limitation section.

Dear reviewer, thank you for your comment.

Answer 3.1: Indeed, we didn’t perform histological analysis on tissues treated because experiments were conducted on ex-vivo tissue with an elevation of only 5 degrees and without irradiation, this kind of elevation cannot permit to see any cellular or histological modification, we added it in the Discussion part (L 534).

Reviewer 4 Report

Dear authors,

thank you for the submission of your work to Cancers.

You describe an interesting concept for transperineal focused ultrasound to induce regional deep hyperthermia in masses in the pelvis.

These are my comments:

Introduction:

In the first paragraph of the introduction, the authors focus very much on local cancer recurrence and difficulties to treat in this setting. The authors may explain further why their novel technique may not be used in previously untreated prostate cancer and/or other pelvic masses (?); as well as the current indications for hyperthermia combined with radiation and how it is currently used (e.g., approved, off-label, compassionate use, clinical trials) and eligibility criteria. As combined radiotherapy and hyperthermia therapy is already in clinical use for prostate cancer, the authors may also summarize its efficacy and toxicity rate (and shorten some other part of the first paragraph).

In the sixth paragraph of the introduction, the authors say they want to present “a system to combine in an adjuvant setting mild regional hyperthermia to radiotherapy as a proof of concept for obtaining sustained biochemical response with low toxicity rates in patients with a macroscopic local recurrence of prostate cancer”. The deduction that a sustained biochemical response may be possible may be discussed in the discussion section and it is speculative.

Please also combine the last two paragraphs and shorten them to clarify your study aims.

Methods&Materials:

Overall, the authors investigate a therapeutic ultrasound application with a transperineal approach. After designing a transperineal focused ultrasound transducer and development of the software, an ex vivo model was used to test the performance of the novel technique. This was an anatomic model mimicking the pelvic window and the prostate bed, consisting of a lamb carcass filled with standard ultrasonic gel and turkey meat. The authors elaborate more on why this model would be specific for prostate (cancer?) or maybe more applicable in pelvic masses (if so, this would also need to be reflected in the introduction, discussion and conclusion).

Discussion:

There is no histological proof for therapeutic efficacy/treatment effects, which should be mentioned in the limitation section of the discussion; and, that in vivo experiments would be needed.

Also, the authors write “While this pilot study represents a proof of concept, for further trials on prostate cancer patients several points must be taken into consideration before clinical application” and the authors might give more information about which points are meant here.

Thanks!

Author Response

Dear authors,

Thank you for the submission of your work to Cancers. You describe an interesting concept for transperineal focused ultrasound to induce regional deep hyperthermia in masses in the pelvis.

These are my comments:

Introduction:

Point 4.1: In the first paragraph of the introduction, the authors focus very much on local cancer recurrence and difficulties to treat in this setting. The authors may explain further why their novel technique may not be used in previously untreated prostate cancer and/or other pelvic masses (?); as well as the current indications for hyperthermia combined with radiation and how it is currently used (e.g., approved, off-label, compassionate use, clinical trials) and eligibility criteria. As combined radiotherapy and hyperthermia therapy is already in clinical use for prostate cancer, the authors may also summarize its efficacy and toxicity rate (and shorten some other part of the first paragraph).

Answer 4.1: Dear reviewer, first thank you for your comment.

In the introduction we decided to focus mainly on radiorecurrent prostate tumors as this is a challenging clinical situation requesting new therapeutic approaches. Nevertheless, we agree that possible applications of this combined treatment may range between definitive radiotherapy and salvage treatments for prostate cancer, or pelvic masses. Of course, under sufficient acoustic access. As suggested, we added a dedicated paragraph on outcome and toxicity results of combined hyperthermia and radiotherapy for prostate cancer (L63). Further details are provided in the discussion. As far as potential clinical applications of hyperthermia outside prostate cancer are concerned, we decided to not develop further this argument as it is not in the scope of the present work (L545).

Point 4.2: In the sixth paragraph of the introduction, the authors say they want to present “a system to combine in an adjuvant setting mild regional hyperthermia to radiotherapy as a proof of concept for obtaining sustained biochemical response with low toxicity rates in patients with a macroscopic local recurrence of prostate cancer”. The deduction that a sustained biochemical response may be possible may be discussed in the discussion section and it is speculative.

Answer 4.2: Indeed, the benefit of combined hyperthermia and radiotherapy in this setting was extrapolated from SBRT but at this stage it remains speculative. We modified the introduction accordingly (L115). We discussed this point in the discussion section (L 551).

Point 4.3: Please also combine the last two paragraphs and shorten them to clarify your study aims.

Answer 4.3: Thank you for your comment, we shortened and combined the last two paragraphs in the introduction to clarify our study aims.

Methods&Materials:

Point 4.4: Overall, the authors investigate a therapeutic ultrasound application with a transperineal approach. After designing a transperineal focused ultrasound transducer and development of the software, an ex vivo model was used to test the performance of the novel technique. This was an anatomic model mimicking the pelvic window and the prostate bed, consisting of a lamb carcass filled with standard ultrasonic gel and turkey meat. The authors elaborate more on why this model would be specific for prostate (cancer?) or maybe more applicable in pelvic masses (if so, this would also need to be reflected in the introduction, discussion and conclusion).

Answer 4.4: We have provided further details under the item 4.1. Thank you for the suggestion.

Discussion:

Point 4.5: There is no histological proof for therapeutic efficacy/treatment effects, which should be mentioned in the limitation section of the discussion; and, that in vivo experiments would be needed.

Answer 4.5. Thank you for the remark, this has also been pointed pointed out by the reviewer 3 (item 3.1). Histological analysis was not considered in this ex vivo study, because we did not anticipate pathologic changes on fresh post-mortem tissue following a temperature elevation of approximately 5 degrees above the room temperature. The need for future in vivo experiments has been added to the discussion. A sentence was added in discussion (L534).

Point 4.6: Also, the authors write “While this pilot study represents a proof of concept, for further trials on prostate cancer patients several points must be taken into consideration before clinical application” and the authors might give more information about which points are meant here.

Answer 4.6: Thank you for pointing this out. Before clinical trials on cancer patients, we need to show that the findings obtained in this study are reproducible on perfused tissue models and of course in vivo. Moreover, we need to analyse and measure the pelvic thermal noise in human using MR thermometry to ensure patient safety. We added this information in the discussion part (L538).

Reviewer 5 Report

This paper describes the design of a trans perineal HIFU transducer operating at 700kHz. The whole design is well described. However I do have a major problem with this paper. This approach seems new but was considered many times in the past. It was always rejected because the access window was too small and the prostate located too deep. Having a major risk of overheating the entrance window. Also with this 700 kHz device the ratio between transducer surface/window and prostate is limited combined with the ≈6 cm penetration depth deep heating of the prostate volume seems unrealistic. This is exaggerated by the extreme cooling of the prostate. It was shown in a clinical study in 2002 that the perfusion of the prostate increased to 50 ml/100gr/min in case of local heating. (Vulpen et al. J Urol 2002, 168, 1597-602) The authors must skip the whole clinical introduction. This is not the time for such clinical rationale and must make an introduction to the problems encountered with prostate heating, ultrasound penetration, thermal modelling, increased perfusion, etc. At least they have to use their models to predict what is going to happen under realistic perfused conditions. The no-perfusion phantom measurements are only useful to calibrate the system and test the ultrasound behaviour, not to test the thermal behaviour.

Author Response

Point 5.1: This paper describes the design of a trans perineal HIFU transducer operating at 700kHz. The whole design is well described. However I do have a major problem with this paper. This approach seems new but was considered many times in the past. It was always rejected because the access window was too small and the prostate located too deep.

Answer 5.1: Dear reviewer, thank you for evaluating the manuscript.

According to the common methodology of literature screening, before developing this prototype device since 2019, we have performed thorough research in Pubmed and Google Scholar about the use of MR-guided therapeutic ultrasound to deliver pelvic or prostate hyperthermia with extracorporeal transperineal approach [search keywords: hyperthermia, HIFU, transperineal, prostate]. There was no item found and this is still the case today. We believe therefore our work is original. We are of course interested in any relevant reference the reviewer could provide to support his statement.

For the sake of precision, we have further found a single report suggesting MR-guided therapeutic ultrasound to deliver prostate hyperthermia via the endorectal route (Salgaonkar et al, Med Phys. 2014 Mar;41(3):033301), using a commercial MR-guided endorectal HIFU ablation array, operating in open loop mode (that is no real time feedback control with MRT during implementation). And a theoretical study without experimental output was published on the targetability of late-stage cervical cancer by magnetic resonance-guided high-intensity focused ultrasound (MRgHIFU)-induced hyperthermia (HT) with a fibroid-dedicated commercial device, as an adjuvant to radiation therapy (RT) (Zhu L et al., Int J Hyperthermia, 2021, 498-501). These references are now cited in the revised manuscript (L96).

Point 5.2: Having a major risk of overheating the entrance window.

Answer 5.2: Our design has taken into account the risk of overheating the entrance window:

  • first, a semi-elastic membrane creates a water flow compartment in contact with the skin, providing convective cooling; the cooling water temperature can be set per need, we used a value of 20°C but this can be further lowered to 10°C or even 4°C ; given the long duration of the sonication, the superficial cooling becomes quite effective also for deeper layers of tissue, see Figure 4, as a temperature gradient is being established.
  • second, the layout of the acoustic elements and the geometry of the device has been engineered to limit this risk, as well as the risk of heating of the adjacent bones.

Our numerical calculations under realistic conditions of perfusion did not bring evidence for a major risk of overheating the entrance window. Of course, in vivo studies are necessary for ultimate validation, while the current manuscript is the first description of the concept and embodiment, with ex vivo experimental results.

Point 5.3: Also with this 700 kHz device the ratio between transducer surface/window and prostate is limited combined with the ≈6 cm penetration depth deep heating of the prostate volume seems unrealistic.

Answer 5.3: Numerical simulations with rather extreme conditions of perfusion cooling of tissue, see Figure 4, demonstrated the delivery of local hyperthermia at 90 mm depth as planned. It is however true that the shape of the heated spot is dependent on the perfusion rate (it becomes narrower hence the treated volume is smaller), this has been included in the revised manuscript.

Point 5.4: This is exaggerated by the extreme cooling of the prostate. It was shown in a clinical study in 2002 that the perfusion of the prostate increased to 50 ml/100gr/min in case of local heating. (Vulpen et al. J Urol 2002, 168, 1597-602)

 Answer 5.4: There are some differences in our approach with respect to the study of Vulpen et al. Here we are limiting the temperature elevation to 5 or 6°C. Also, we are using a closed loop temperature controller that is supposed to react in real time to increased perfusion during hyperthermia. Other publications give far lower values for perfusion in pelvic tumours (Tilly et al., Int J Hyperthermia, 2001, 172-188). Aiming to accommodate the reviewer’s remark, we have performed new simulations considering a temperature-dependent perfusion as a quadratic model where the interval of perfusion values was set from 0.0011 s-1 to 0.0083 s-1 (L229). The highest value was taken from Vuelpen et 2002 after converting the physical units to the international system. Please find new Figure 4 (L408).

Point 5.5: The authors must skip the whole clinical introduction. This is not the time for such clinical rationale and must make an introduction to the problems encountered with prostate heating, ultrasound penetration, thermal modelling, increased perfusion, etc.

Answer 5.5: We think that the clinical introduction is important in this article to show readers the value of this kind of technique if this could be used in clinical routine. However, we added in the introduction more details about the difficulties of prostate heating (L110).

Point 5.6: At least they have to use their models to predict what is going to happen under realistic perfused conditions.

Answer 5.6: We performed numerical simulation with different perfusion conditions, added in the Methods and Result, also please find the new Figure 4 (L408). We believe this is improving the manuscript and makes it more convincing, thank you for the suggestion.

Point 5.7: The no-perfusion phantom measurements are only useful to calibrate the system and test the ultrasound behaviour, not to test the thermal behaviour.

Answer 5.7: Within the scope of this initial description of the concept and embodiment, the manuscript needs also to fit a maximum length. We reported therefore the design, manufacturing and experimental results in non-perfused tissue with closed looped automatic temperature control. One model of perfused phantom has been described by our group (Lorton et al, PMID: 32990101) and is actually being integrated with the described device, for a future report.  We would like to highlight the closed loop temperature feedback that should compensate in real time for perfusion effects.

Round 2

Reviewer 4 Report

Dear Authors,

thank you for your additional work addressing the comments.